# The Gut-Brain Axis in Autism Spectrum Disorder: A Focus on the Metalloproteases ADAM10 and ADAM17

**DOI:** 10.3390/ijms22010118

**Published:** 2020-12-24

**Authors:** Yuanpeng Zheng, Tessa A. Verhoeff, Paula Perez Pardo, Johan Garssen, Aletta D. Kraneveld

**Affiliations:** 1Division of Pharmacology, Utrecht Institute for Pharmaceutical Sciences, Faculty of Science, Utrecht University, 3584CG Utrecht, The Netherlands; y.zheng@uu.nl (Y.Z.); t.a.verhoeff@students.uu.nl (T.A.V.); p.perezpardo@uu.nl (P.P.P.); j.garssen@uu.nl (J.G.); 2Global Centre of Excellence Immunology, Danone Nutricia Research B.V., 3584CT Utrecht, The Netherlands

**Keywords:** Autism Spectrum Disorder (ASD), gut-brain axis, neuroinflammation, metalloproteases, A Disintegrin And Metalloprotease (ADAM), ADAM10, ADAM17, ectodomain shedding

## Abstract

Autism Spectrum Disorder (ASD) is a spectrum of disorders that are characterized by problems in social interaction and repetitive behavior. The disease is thought to develop from changes in brain development at an early age, although the exact mechanisms are not known yet. In addition, a significant number of people with ASD develop problems in the intestinal tract. A Disintegrin And Metalloproteases (ADAMs) include a group of enzymes that are able to cleave membrane-bound proteins. ADAM10 and ADAM17 are two members of this family that are able to cleave protein substrates involved in ASD pathogenesis, such as specific proteins important for synapse formation, axon signaling and neuroinflammation. All these pathological mechanisms are involved in ASD. Besides the brain, ADAM10 and ADAM17 are also highly expressed in the intestines. ADAM10 and ADAM17 have implications in pathways that regulate gut permeability, homeostasis and inflammation. These metalloproteases might be involved in microbiota-gut–brain axis interactions in ASD through the regulation of immune and inflammatory responses in the intestinal tract. In this review, the potential roles of ADAM10 and ADAM17 in the pathology of ASD and as targets for new therapies will be discussed, with a focus on the gut–brain axis.

## 1. Introduction

Autism Spectrum Disorder (ASD) is a spectrum of neurodevelopmental disorders that are generally diagnosed early in life and might persist across the whole lifespan. It is mainly characterized by a deficiency in social interactions and communication and the presence of specific stereotyped behaviors [1]. The prevalence of ASD is approximately 1.5% in developed countries, as determined in 2012 by the World Health Organization [2], and predominantly affects males [3]. Globally, the incidence of ASD has shown a 35-fold increase (1 in 59 children in the USA and 1 in 89 in the EU are affected) compared to the 60s and 70s (Centers for Disease Control and Prevention Data & Statistics on Autism Spectrum Disorder, 2019 and Autism Spectrum Disorders in the European Union, 2018). A diagnosis is performed using a behavioral assessment due to the absence of reliable biomarkers.

Although the ASD pathogenesis remains elusive, it is thought that it results from early altered brain development and neural reorganization [4,5]. Some clinical aspects are found in large groups of people diagnosed with ASD, such as altered neuronal connectivity, increased synaptic density [6], neuroinflammation [7], microbiota dysbiosis [8,9,10], dysregulated immune responses [11,12,13,14,15] and gastrointestinal abnormalities [16,17,18]. In addition, the most replicated neuroanatomical finding in infants and the early childhoods of people diagnosed with ASD is an enlarged brain volume, called macrocephaly or megalocephaly [19,20]. Genetically, it is estimated that 400–1000 genes are involved in ASD susceptibility, and it is thought that there are more that have and will be discovered in the near future [21]. The estimated heritability of ASD was 83% in a recent population-based cohort of children born in Sweden [22], and Bailey et al. reported a similar result [23]. Among these genetic predispositions, the membrane-bound synaptic genes for the Amyloid Precursor Protein (APP), Neural glial-related Cell Adhesion Molecule (NrCAM), Neuroligins (NLGNs), Neurexins (NRXNs) and Protocadherins (PCDHs) were widely identified as candidate genes of ASD [24,25,26,27,28], as their deficiency in mice led to ASD-like behaviors, such as deficits in spatial memory and learning, increased repetitive and stereotype grooming behaviors and compromised social interactions [29,30,31]. Interestingly, the amount of these ASD-related proteins in the membrane is controlled by the metalloproteases ADAM10 and ADAM17 through proteolytically cleaving these transmembrane proteins by which ADAM10 and ADAM17 might be involved in ASD pathogenesis [32,33,34,35,36].

A Disintegrin And Metalloproteases (ADAMs) are a subgroup of the metzincin family of metalloproteases, which also consists of Matrix Metalloproteases (MMPs), ADAMs with thrombospondin motifs (ADAMTSs) and Snake Venom Metalloproteases (SVMPs) [37,38]. ADAMs are ubiquitously expressed and are able to regulate sperm–egg interactions, cell proliferation, differentiation, migration and cell fate determination [39]. ADAM10 and ADAM17 are the two most investigated members of the ADAM family. Interestingly, both ADAM10 and ADAM17 are highly expressed in the brain, as well as the intestines. In the Central Nervous System (CNS), ADAM10 and ADAM17 are able to regulate axon guidance and synaptic functions through controlling the cleavage of synaptic proteins, such as APP, NrCAM, NLGNs, NRXNs and PCDHs. Importantly, ADAM10 plays a vital role in synaptic pruning by cleaving the chemokine fractalkine (CX3CL1) that binds to its receptor CX3C chemokine receptor 1 (CX3CR1) to induce microglia-mediated synapse elimination [40]. ADAM17 also regulates neuroinflammation, attributing to its capacity of converting membrane-bound Tumor necrosis factor-α (TNF-α) into a soluble form that recognizes the TNF-α receptors I and II and, consequently, triggers inflammatory responses. In the intestinal tract, ADAM17 can regulate intestinal inflammation, intestinal barrier permeability and inflammatory responses by cleaving several cytokines, such as TNF-α and lymphotoxins [41,42,43,44]. Moreover, ADAM10 can control the intestinal permeability by cleaving the transmembrane proteins Notch [45,46,47] and E-cadherin [48]. However, there are only a few reports available that elucidate the roles and functions of ADAM10 and ADAM17 in the gut-to-brain pathology of ASD.

Taken together, ADAM10 and ADAM17 regulate synaptic functions, neuroinflammation and brain development, as well as intestinal barrier functions, inflammation and immunity, which are involved in the pathogenesis of ASD. In this review, we aim to elaborate on the potential role of ADAM10 and ADAM17 in the pathogenesis of ASD with a major focus on the gut–brain axis.

## 2. Structure of Metalloproteases

The metzincin family of metalloproteases has four family members: ADAMs, MMPs, ADAMTSs and SVMPs [37,38]. They are called metzincins for the conserved Met residue at the active site and the use of a zinc ion in the enzymatic reaction. Collectively, the metalloproteases are able to degrade all extracellular structures. These family members have some corresponding protein domains (Figure 1). All members start with a signal peptide at their N-terminal that allows them to be located at the secretory pathway. Immediately following the signal peptide is a pro-domain. This ensures enzyme latency until cleaved by pro-protein convertases [49]. After the pro-domain, all members of the metzincin family contain a metalloprotease domain, which holds its catalytic activity. After this, some major structure differences are found between members. For ADAMs, SVMPs and ADAMTSs, the metalloprotease domain is followed by a Disintegrin domain and then a cysteine-rich domain. For the ADAMTS, the cysteine-rich domain follows the Thrombospondin region. The ADAMs contain an epidermal growth factor (EGF)-like domain, followed by a transmembrane and cytoplasmic domain. SVMP and ADAMTS are soluble proteins. Membrane-type MMP and MMP contain a Hemapexin domain after their metalloprotease domain, necessary for substrate selectivity and for binding with Tissue Inhibitors of Metalloproteinases (TIMPs), the main MMP inhibitor [37], and the membrane-type matrix metalloproteinase (MT-MMP) contains a transmembrane region with a cytoplasmic tail [37,50,51].

In this review, the focus will be on two types of ADAMs: ADAM10 and ADAM17. The MMPs, MT-MMPs, ADAMTSs and SVMPs are beyond the scope of this review. Little is known about the possible role of MT-MMPs, ADAMTSs and SVMPs in ASD, however, it should be noted that we do not rule out that MMPs, such as MMP9, might also play a role in the pathogenesis of ASD [52,53].

### ADAMs

ADAMs are a family of type I transmembrane proteins characterized mainly by their ability to cleave membrane-bound proteins at their extracellular domain. The cleavage generates a soluble protein fraction in the extracellular space, a process called “ectodomain shedding”. This will influence the signaling pathways of other cells by decreasing the amount of membrane-bound receptors or by increasing the amount of soluble ligands [50]. ADAMs are proteins of approximately 750 amino acids that contain several structurally conserved domains, which determine its biological function. The metalloprotease domain can contain a catalytic site with a zinc-binding motif, which is mediated by three histamine residues (HEXGHXXGXXHD) [54]. Around 24 ADAMs have been identified in humans, of which only 12 contain the metalloprotease domain with the active zinc-binding site [55,56]. The biological function of these proteolytically active ADAMs (ADAM8, 9, 10, 12, 15, 17, 19, 20, 21, 28, 30 and 33) is determined by their substrates and includes sperm–egg interactions, cell migration, axon guidance, inflammation and cell fate determination [39]. In addition, ADAMs have been implicated in different pathologies, such as cancer [57], inflammation [58] and Alzheimer’s Disease (AD) [59,60].

## 3. ADAM10 in the Central and Enteric Nervous Systems

The most extensively studied member of the ADAMs family is ADAM10. There are more than a hundred substrates cleaved by ADAM10 in the CNS [35], and its expression, maturation and substrates selectivity are regulated by the TspanC8 subfamily of tetraspanin, consisting of Tspan5, 10, 14, 15, 17 and 33 [61,62,63], as different Tspan-ADAM10 complexes might adopt different conformations and spaces to their substrates [62,64,65]. ADAM10 is ubiquitously expressed in the brain [66], where it is located at the synapse and in synaptic vesicles and functions as a sheddase of other synaptic proteins [67], which makes ADAM10 able to control CNS processes, such as development, synaptogenesis and axon targeting. In addition, ADAM10 is expressed in the intestinal tract [68]. In Section 5, the role of ADAM10 in the intestines is discussed. Table 1 lists the major ADAM10 substrates in the CNS that will be discussed below.

### 3.1. Amyloid-β Precursor Protein (APP)

APP is a transmembrane protein involved in cell adhesion and neurite pruning [69,70]. It consists of an extracellular N-terminus domain, a transmembrane region and a C-terminus, intracellular domain [71]. APP can be proteolytically cleaved by a group of secretases: α-, β- and γ-secretases (Figure 2). The β-site APP cleaving enzyme 1 (BACE1) and γ-secretases induce the amyloidogenic pathway, where APP is cleaved extracellularly to create a soluble fraction, sAPPβ, the main component of AD plaques, Amyloid β-peptide (Aβ) and Amyloid Precursor Intracellular Domain (AICD) [72]. α-Secretases activate the nonamyloidogenic pathway by creating the soluble fractions sAPPα, P3 and AICD. ADAM10 is the main α-secretase of APP in the nervous system [32]. ADAM10 cleaves APP in the Aβ domain, which inhibits the formation of the pathological plaques that cause AD and, consequently, creates sAPPα instead. Therefore, the nonamyloidogenic pathway of APP is thought to serve a neuroprotective function at this point [72]. Interestingly, studies show that while ADAM10 is the constitutive secretase, ADAM17 is the stimulatory secretase of APP [32].

Recent studies have started to investigate the roles of ADAM10 and ADAM17 in neurodevelopmental diseases. Of all ADAM10 substrates, APP is the most described in the context of ASD. It has been shown that there is an increase of sAPPα levels in the plasma of children diagnosed with severe ASD aged between 5–17 years [28,73]. In the fragile X mental retardation 1 knockout (Fmr1 KO) mouse model of Fragile X Syndrome, significantly increased levels of both sAPPα and ADAM10 are found at postnatal day 21 [74] but not in adulthood. These findings are in-line with the concept that ASD is a result of early altered brain development, as the prenatal and perinatal period is most critical for synaptogenesis. In parallel with these main findings, Westmark et al. found that genetically decreasing APP and Aβ levels are able to ameliorate the autistic phenotype in adult Fmr1 KO mice [75]. Additionally, Lahiri et al. hypothesized that increased sAPPα levels can activate neuroprotective pathways and microglia, which result in neuronal overgrowth and neuroinflammation, leading to an increased brain volume that is also observed in ASD [19,20,76].

Importantly, the N-terminal of APP (N-APP) is a ligand of Death receptor 6 (DR6), which is highly expressed in oligodendrocytes and neurons. N-APP is a cleavage product of sAPPβ by a still undetermined mechanism, and the specific cleavage site is unknown. The binding of N-APP to DR6 triggers neuronal death via caspase pathways in vitro and in vivo, and consequently, a role for N-APP/DR6 in neurodegeneration has been proposed [70]. Furthermore, DR6 negatively regulates oligodendrocyte survival, maturation and myelination, which is related to microglia activation, phagocytosis and neuroinflammation [77,78,79]. Colombo et al. demonstrated that the DR6 of Schwann cells (SCs) negatively regulates the myelination of the Peripheral Nervous System (PNS) and that DR6 KO mice showed precocious myelination in the PNS [80]. SCs underlie the sheath of most of peripheral nerves and regulate the myelination of the nervus vagus in the PNS [81,82,83]. Vagal stimulation is recognizably involved in ASD development. Sgritta et al. demonstrated that Lactobacillus reuteri (L. reuteri) rescued social behaviors in ASD mice (SH3 and multiple ankyrin repeated domains 3B KO mice) but not in vagotomized mice. These findings indicate that L. reuteri might ameliorate ASD-like behavior in the ventral tegmental area of ASD mice in a vagus nerve-dependent manner [84,85]. In addition, Jin et al. proposed transcutaneous vagus nerve stimulation is a promising treatment for ASD, but the exact mechanism is not clear [86]. Recently, DR6 cleavage was decreased by 50% in ADAM10-deficient murine neurons, and consequently, it is a substrate of ADAM10 [80]. However, the potential role of ADAM10-mediated cleavages of APP and its receptor DR6 in the pathology and treatment of ASD is barely investigated.

Overall, ADAM10 and, possibly, ADAM17 are able to regulate APP shedding and create sAPPα fractions at the expense of sAPPβ. As elevated sAPPα levels are found in the plasma of children with severe ASD behavior, it is probable that ADAM10/17-mediated APP shedding contributes to the development of disturbed brain development in ASD. More studies are needed to elucidate the specific mechanisms of APP shedding in ASD.

### 3.2. Neuroligins (NLGNs) and Neurexins (NRXNs)

NLGNs are synaptogenic adhesion proteins located at the post-synapse that trans-synaptically binds to the presynaptic partner NRXNs to form a NRXN/NLGN complex, necessary for efficient neurotransmission. These two proteins recruit key synaptic proteins, such as scaffolding proteins and neurotransmitter receptors, after the initial contact of an axon with its target cell. Therefore, they are essential for synaptic formation, maturation and differentiation [87,88,89]. There are five types of NLGNs (NLGN1, NLGN2, NLGN3, NLGN4 and NLGN4Y) and three NRXNs (NRXN1, NRXN2 and NRXN3) in the human genome. They are involved in ASD pathogenesis. Among the NLGNs, NLGN3 is the strongest candidate, followed by NLGN1, and of the NRXNs, NRXN1 is the strongest candidate [90]. Loss-of-function variants of NRXN1 have been found in ASD patients [24,26,27]. Interestingly, variants of the other two types, NRXN2 and NRXN3, are much rarer. Furthermore, NLGN-1 KO mice display deficits in spatial memory and learning and an increased repetitive, stereotypical grooming behavior, which is accompanied by a reduced ratio of NMDA to the α-amino-3-hydroxy-5-methyl-4-isoxazolepropionic acid receptor (AMPA) at the corticostriatal synapses [29]. NLGN-1 is shed by ADAM10 for 83% and NLGN-3 for 62%, as described in ADAM10-deficient neurons (Table 1) [35]. As the other types of NLGNs are not cleaved by ADAM10, we will not discuss them further. NLGN-1 is found exclusively on glutamatergic neurons, whereas NLGN-3 is found on both glutamatergic and GABAergic synapses [91,92]. The proteolytic cleavage by ADAM10 of membrane-bound NLGN-1 increases by either N-methyl-D-aspartate (NMDA) receptor activation or by binding to the secreted form of NRXNs [92]. Interestingly, the secreted form of NRXN2 and NRXN3 may also be generated by ADAM10 or ADAM17 [33,34]. A recent study discovered that NRXN1 is primarily cleaved by ADAM10 in hippocampal neurons [93].

Only limited reports are available on the role of intestinal NLGN-1, NLGN-3 and NRXN. NLGN-3 is expressed in the enteric nervous system as well, and gastrointestinal dysfunction is found in people and mice with a R451C missense mutation in this NLGN-3 that have an ASD phenotype [94]. Very recently, it was shown that ASD-associated NLGN-3 mutations, as well as NLGN-3 KO mice, have more cecal Nitric Oxide (NO)-producing neurons and more activated enteric macrophages [95]. These phenomena might explain the presence of intestinal symptoms in these NLGN-3-deficienct mice [94,96] and ASD patients [97,98], such as a disturbed intestinal transit and intestinal inflammation. Additionally, NLGN-1 and NRXN are shown to be important for the development of the enteric nervous system in rats [99]. It is rather speculative to link the enhanced intestinal ADAM10/17 expression to the loss of NLGN-1, NLGN-3 or NRXNs and ASD-related intestinal dysfunction.

Taken together, ADAM10 cleaves NLGN-1, NLGN-3 and, possibly, NRXNs. Following this, it can be hypothesized that increased ADAM10 levels or activity will lead to an increased cleavage of NLGN-1, NLGN-3 and NRXNs, therefore terminating their function at the synapse and disrupting the neurotransmissions. Disturbances in neurotransmissions are also described in people diagnosed with ASD. It has been shown that a decreased glutamate concentration in the striatum correlates with the severity of social ASD symptoms, suggesting that glutamate/γ-aminobutyric acid (GABA) abnormalities in the corticostriatal circuitry may contribute to ASD development [100]. Moreover, it was shown that NLGN-1 shedding has a negative effect on NRXN1 stability [101], which also seems to be primarily cleaved by ADAM10 [93]. In conclusion, ADAM10 is a major regulator of synaptic functions of NLGN/NRXN complexes, and the loss of NLGN-1 and NLGN-3 in mouse models induces autistic-like phenotypes [29,102,103], which highlights a potential connection between ASD and ADAM10-mediated cleavages of NLGN/NRXN complexes.

### 3.3. Protocadherins (PCDHs)

Cadherin is a family of calcium-dependent cell adhesion proteins consisting of three family members: classic cadherins, desmosomal cadherins and PCDHs [104]. PCDHs are mainly expressed in the developing CNS [105]. One member of this family that is an ADAM10 substrate is PCDH9 [35]. The conditional deletion of ADAM10 in primary neuron cultures resulted in a 71% reduced shedding of PCDH9 [35], which has been proposed to play a role in synaptogenesis [106]. It remains unclear what the effect of ADAM10 shedding is on PCDH9 functions. In PCDH9-deficient mice, long-term social and object recognition deficits were determined [30] but without any changes in perception, sociability or fear memory. Furthermore, the PCDH9 KO mice showed impairments in sensorimotor development and structural changes in layers of sensory cortices [30]. The behavioral dysfunctions in PCDH9 KO mice are also presented in ASD. Moreover, Copy Number Variations (CNV), including deletion, duplication, translocation and inversion, of the PCDH9 gene have been found in patients with ASD [24]. Additionally, another member of PCDHs, PCDH8, has also been identified as a substrate of ADAM10 [35] and regarded as a candidate ASD gene in Caucasian females [107]. Furthermore, Breuillard et al. demonstrated, although with a limited sample size, that PCDH19 genetic defects frequently emerged in female ASD children with epilepsy and intellectual disability [108]. Obviously, PCDHs play an important role in ASD pathogenesis, as it seems that ADAM10 is the main sheddase of PCDH8 and PCDH9 [108]. Indeed, the role of ADAM in the loss of function of PCDHs is not yet investigated in ASD. To elucidate this connection further, more research will be necessary.

### 3.4. Neural Glial-Related Cell Adhesion Molecules (NrCAM)

NrCAM is part of the L1 family of cell adhesion molecules of immunoglobulin superfamily (IgCAMs), and a cell adhesion molecule [109]. NrCAM is involved in brain development, where it controls dendritic spine densities, axon guidance and targeting and neurite outgrowth [110,111,112]. NrCAM was shown to be an ADAM10 substrate, where the deletion of ADAM10 in primary neurons reduces NrCAM shedding by 66% [35]. In addition, Brummer et al. recently showed that ADAM17 deletion in primary neurons did not alter NrCAM proteolysis and that ADAM10 controls NrCAM cell surface expression levels and NrCAM-dependent neurite outgrowth in vitro [36]. Furthermore, mice with a conditional ADAM10 knockout in neurons showed increased cell surface expression levels of NrCAM [35,36] and a reduced number of dendritic spines [113], whereas NrCAM knockout mice show increased dendritic spine densities. As NrCAM is a member of the Sema3F complex that mediates spine retraction [114], increased NrCAM surface levels would be expected to decrease spine density. A potential role for soluble NrCAM in axon signaling becomes apparent in NrCAM-deficient [115] and ADAM10-deficient mice [35]. Both mouse models present with axonal targeting deficits within their olfactory bulbs with axons overshooting their marks. This would suggest a functional role for the soluble fraction of NrCAM generated by ADAM10 cleavage, although more research is needed to obtain more information about this role. Currently, no studies report on the possible peripheral/intestinal role of NrCAM in relation to ASD.

In NrCAM-deficient male mice, the loss of NrCAM leads to ASD-related behavioral alterations in sociability, acquisition of a spatial task and reversal learning [31]. Marui et al. identified that seven single-nucleotide polymorphisms (SNPs) within the NrCAM gene are associated with ASD in Japanese children [25]. Bonora et al. also detected in 48 unrelated individuals with ASD several polymorphisms in the promoter and untranslated region of NrCAM and suggested that a reduction in the expression of this gene might be involved in ASD susceptibility [116]. In contrast, Hutcheson et al. showed that there is no association between ASD susceptibility and the NrCAM gene in the subsets of chromosome 7-linked families [117]. In order to further explore the relation between ASD and NrCAM, and the possible role of the ADAM10-induced cleavage of NrCAM, more studies are needed.

### 3.5. Fractakine (CX3CL1)

CX3CL1 is the only member of the δ subfamily of chemokines that is constitutively and abundantly expressed in the brain—specifically, in glial cells and neurons [118]. CX3CL1 recognizes its receptor CX3CR1, which is exclusively expressed in the microglia [119,120]. CX3CL1–CX3CR1 signaling is necessary for the immune response, neuroinflammation, synaptic pruning and brain development through maintaining the phagocytic function of the microglia [40,119]. Moreover, CX3CL1 was identified as a substrate of ADAM10 and cleaved by ADAM10 to generate a soluble CX3CL1 that acts as a ligand of CX3CR1 in the brain [35,121,122]. Currently, there is little known about the link between the ADAM10-mediated cleavage of CX3CL1, microglial phagocytosis and ASD. However, it has been shown that CX3CR1 KO mice exhibit a deficiency in microglia engulfment and, consequently, show an increased density of dendritic spines and immature synapses, as well as a synaptic pruning deficiency [119]. Furthermore, there is a defect in synaptic elimination in both CX3CL1 KO mice and CX3CR1 KO mice, which was phenocopied after ADAM10 inhibition in wild-type mice [40]. Insufficient synaptic elimination is a cause of hyperconnectivity in the brain, which is related to the macrocephaly found in ASD patients. Additionally, Rogers et al. demonstrated that CX3CR1 KO mice show impairments in synaptic plasticity and cognitive function [123], which are symptoms of neurodevelopmental diseases, including ASD.

There are limited reports available that discuss the contribution of ADAM10-mediated cleavage of CX3CL1 to the intestinal disturbances found in ASD. Interestingly, the CX3CL1–CX3CR1 levels are critical for the sex differences in high-fat food-induced obesity. For instance, females are much more resistant to diet-induced obesity due to the higher expression levels of CX3CL1 than in males. Strengthening this point, female CX3CR1 KO mice phenocopied “male-like” microglial activation and increased their susceptibility to diet-induced obesity. Conversely, increasing the CX3CL1 levels in the male mice brain converted them to a “female-like” metabolic phenotype with a decrease of microglial activation and weight gain [124]. These sex differences may be involved with gender differences in ASD. However, more research is necessary.

Taken together, the important role that CX3CL1–CX3CR1 signaling plays in the phagocytic function of microglia and, consequently, synaptic pruning emphasizes the potential of this ADAM10 substrate to contribute to ASD pathology. Importantly, more research will be needed to further explore alterations in CX3CL1 expression in people diagnosed with ASD and its contribution to the disease.

## 4. ADAM17 in the Central Nervous System

ADAM17 is one of most extensively studied member of the ADAMs family and is ubiquitously expressed in all tissues and cell types. Similar to the regulation of ADAM10 by TspanC8, the selectivity or specificity of substrates in ADAM17 is regulated by the iRhoms subfamily, which is one of the rhomboid superfamilies of intramembrane proteases and consists of iRhom 1 and iRhom 2 [125,126]. In addition, iRhoms also regulate ADAM17 enzymatic maturation from its exit from the endoplasmic reticulum to the membrane [125,127]. ADAM17 was first discovered as the enzyme responsible for the proteolytic cleavage of TNF-α, and therefore, ADAM17 was originally called the TNF-α Converting Enzyme [128]. Currently, it is known that ADAM17 is responsible for the shedding of over 90 substrates. Some substrates of ADAM17, such as TNF-α, Tumor necrosis factor-α receptor, Interleukin-6 receptor (IL6-R) and Triggering receptor expressed in myeloid cells-2 (TREM2), are involved in the onset of immune responses and neuroinflammation [129]. The influence of neuroinflammation in ASD has been recently reviewed [7]. Additionally, it was shown that ADAM17 expression levels increased with age in juvenile people diagnosed with ASD [130], although this was reported in relation to the elevation of sAPPα and not with respect to neuroinflammation, inflammation and immune responses. We will discuss the proteolytic cleavages of these substrates, focusing on neuroinflammation and immunity in the CNS and their role in ASD below. More information involving ADAM17 in the intestinal tract will be provided in Section 5.

### 4.1. Tumor Necrosis Factor-α (TNF-α)

ADAM17 is the main protease of TNFα, a proinflammatory cytokine that can elicit its proinflammatory potential only after being proteolytically released from the cell surface [128,131]. Although deletion of the ADAM17 gene abolishes TNF-α shedding by 90% in ADAM17^−/−^ macrophages and neutrophils, it seems that there are other proteases, amongst others MMP7 and proteinase 3, that are also able of proteolytically cleaving TNF-α [131]. In addition, ADAM10 was identified as a major sheddase when ADAM17 is deficient in fibroblasts, indicating that there is a compensation between ADAM10 and ADAM17 [132]. These reports demonstrated that these two metalloproteases are important in TNF-α cleavage. It was shown that increased levels of TNF-α have been found in the cerebellum and hippocampus in a murine model for ASD induced by Valproic Acid (VPA) [133] and in the brain cortex of people diagnosed with ASD [134].

Together, there is an increase in proinflammatory cytokines in the brains of people diagnosed with ASD, and it seems to be that decreasing the levels of these specific cytokines has a beneficial effect on the disturbed sociability. As ADAM17 has been shown to be the main protease that controls TNF-α shedding from the cell membrane, an interesting connection between ADAM17-mediated TNF-α cleavage, neuroinflammation and ASD remains to be elucidated.

### 4.2. Interleukin-6 Receptor (IL6-R)

IL6-R is a known substrate of ADAM17. The cytokine IL-6 has both pro- and anti-inflammatory properties, which is determined by the receptor signaling type [129,135]. Signaling via the membrane-bound IL-6R is called classic signaling. This type of signaling can only occur on cell types that express IL-6R on their surface and is thus limited to hepatocytes and several leukocyte subsets and results in an anti-inflammatory response [136,137]. Signaling via soluble forms of the IL-6R, which is called *trans*-signaling, can occur on all cell types, because the IL-6/soluble IL-6R (sIL-6R) complex can directly bind to and activate the ubiquitously expressed glycoprotein 130 (gp130). The *trans*-signaling accounts mainly for the proinflammatory properties of IL-6 [137,138]. Interestingly, research in transgenic mice showed that the proteolytic cleavage of IL-6R to create a soluble form is carried out by both ADAM17 and ADAM10 [139] and that approximately 85% of sIL6-R is a result of a proteolytic cleavage in vivo [140]. Increased levels of IL-6 in the brain have been found in a murine model for ASD [133] and in the brain cortex of ASD patients [134]. Moreover, it has been shown that overexpressing IL-6 in the mouse brain mediates neuronal circuitry imbalances and induces ASD-like behavior [141]. Furthermore, one study showed that blocking the *trans*-signaling pathway of IL-6 led to improved social behavior in a murine ASD model by continuously infusing the IL-6 *trans*-signaling blocker sgp130Fc protein [142]. As ADAM17 and ADAM10 are capable of proteolytically cleaving IL6-R to create the soluble IL6/sIL6-R complex, it will be of interest to investigate whether reducing the levels of these metalloproteases and, consequently, the level of sIL6-R will lead to reduced ASD-like behavior [142].

### 4.3. Triggering Receptor Expressed in Myeloid Cells-2 (TREM2)

ADAM17 is the main protease shedding TREM2 under steady-state conditions [143]. Whether ADAM10 has a similar effect on TREM2 is still under debate [143]. TREM2 is a type I transmembrane protein and exclusively expressed by the microglia [144,145]. TREM2 deficiency in mice decreases the number of microglia and activated microglia in the hippocampus and increases the synaptic and spine density [145]. This may be involved with the increased expression of ligands of TREM2 induced by apoptotic neurons [146]. Furthermore, it is demonstrated that TREM2 is essential for initiating microglia-dependent synaptic pruning during early brain development [145]. Synaptic pruning is essential to remove synapses and keep normal brain connectivity during brain development. The shedding of TREM2 of microglial cells by ADAM17 might result in reduced synaptic pruning and associated neuronal overgrowth. The shedding of TREM2 results in a soluble fraction, sTREM2. Recently, Zhong et al. demonstrated that sTREM2 is able to activate the microglia and increases neuroinflammatory responses both in vitro and in vivo [147]; however, by which mechanisms remains to be elucidated.

Reports on the connection between TREM2 and ASD are currently scarce. Trem2-deficient mice display increased synaptic density, enhanced excitatory neurotransmission and reduced social and repetitive behaviors [145]. Additionally, in the post-mortem brain tissue of ASD patients, reduced TREM2 levels were found in the age group of 5–23 [145]. Furthermore, TREM2 protein levels of brain tissue were inversely correlated to the Autism Diagnostic Interview-Revised score [145]. However, the levels of sTREM2 were not determined in this study. In general, reduced TREM2 could result in an altered control of microglial pruning, and consequently, this would lead to increased synaptic density. In ASD patients, synaptic density is increased on apical dendrites of pyramidal neurons from cortical layer 2 in the frontal, temporal and parietal lobes and layer 5 only in the temporal lobe [6]. Therefore, the relationship between the lower TREM2 levels and increased synaptic density in ASD patients seems plausible.

Taken together, it is difficult to determine the contribution of ADAM17 or ADAM10-mediated shedding of TREM2 in people diagnosed with ASD. The decreased levels of TREM2 in juvenile ASD could be related to either lower baseline TREM2 protein levels or to increased shedding of the protein by proteases. This can be determined by measuring the sTREM2 levels in the same post-mortem tissue.

## 5. ADAM10 and ADAM17 in the Gut–Immune–Brain Axis

The contribution of the intestines to ASD pathogenesis remains a field of active research. Forty-six point eight percent of people diagnosed with ASD present with at least one intestinal symptom [148], such as constipation [149] and diarrhea [16]. Furthermore, a meta-analysis revealed that children with ASD show four times more intestinal symptoms than controls [150]. The gut–brain axis refers to the bidirectional interaction between these organs [151]. Alterations in this pathway could lead to the increased permeability of both the intestinal and brain barriers. Research emphasizing the role of the intestines show that, in the post-mortem duodenal tissue of people diagnosed with ASD, elevated levels of pore-forming proteins and decreased levels of barrier-forming proteins were found in the tight junction of the intestinal epithelium [53]. These findings suggest a “leaky gut”, which could lead to more circulating bacterial metabolites in the blood of people diagnosed with ASD and activation of the immune system associated with an enhanced cytokine response [152,153,154]. Entering the brain by circulating cytokines and bacterial metabolites via blood is regulated by the Blood–Brain Barrier (BBB). The BBB allows the selective entrance of compounds to the brain through the expression of receptors and transporters, which are necessary for maintaining brain homeostasis [155,156]. Disruption of the BBB—for example, by chronic systemic inflammation—will increase the permeability and allow cytokines and bacterial metabolites to enter the brain directly. In turn, this could result in neuroinflammation and neuronal dysfunction [156,157]. There are limited studies investigating the effect of ADAM10 and ADAM17 on the immune system, the intestinal homeostasis, intestinal microbiota, in intestinal inflammation and epithelial and endothelial (BBB) barrier functions. These studies are discussed below.

### 5.1. ADAM10 and ADAM17 and Blood–Brain Barrier Permeability

There is little known regarding the effects of ADAM10 or ADAM17 on the BBB permeability. Schulz et al. demonstrated ADAM10 increases the endothelial permeability by specifically cleaving Vascular Endothelial Cadherin (VE-cadherin) in human umbilical vein endothelial cells [158]. The low-density lipoprotein receptor-related protein 1 (LRP1) prevents the endocytic transport of Aβ [159,160]. LRP1 is located at the abluminal surface of the brain endothelium, by which Aβ is then released into the systematic circulation. ADAM10 KO and ADAM10 inhibition facilitate the clearance of Aβ in the brain through decreasing the proteolytic cleavage of LRP1 by ADAM10 in mice and in human brain microvessel endothelial cells [161]. It seems plausible that ADAM10 plays an important role in changing the BBB permeability through the proteolytic cleavage of junction and transporter proteins in the BBB.

### 5.2. ADAM10 in the Intestinal Tract

ADAM10 is widely expressed in intestinal epithelial cells [68] and involved in modulating the intestinal permeability by targeting its substrates Notch and E-cadherin [48]. Research in conditional ADAM10-deficient mice points out that, when ADAM10 is deleted in intestinal cells, there is an early lethality caused by altered intestinal morphology and changes in cell differentiation [45]. Furthermore, it has been shown that the intestinal morphology changes were due to the loss of Notch receptor signaling caused by the shedding of ADAM10 [45]. The Notch receptor is a recognized substrate of ADAM10 that is ubiquitously expressed in all epithelial cell types. The Notch receptor determines the intestinal stem cell fate and controls intestinal homeostasis [162]. An increase of cleaved Notch-1 decreases the transepithelial electrical resistance, indicative for a reduced intestinal barrier function, and tight junction protein Claudin-5 expression in Caco-2 cells. In addition, the levels of cleaved Notch-1 are increased in the colonic epithelium of patients with Crohn’s disease [47]. E-cadherin is one of the most important junction molecules involved in maintenance of the intestinal epithelial integrity. It was demonstrated that E-cadherin is specifically cleaved by ADAM10 in mouse embryonical fibroblasts and by the absence of soluble E-cadherin in ADAM10-deficient mice [48]. Taken together, these findings suggest that the ADAM10-mediated shedding of Notch receptor and E-cadherin downregulates epithelial cell migration and adhesion and influences intestinal barrier function.

Finally, there might be a role for ADAM10 (possibly, also, ADAM17) in cleaving APP in relation to weight gain, as children diagnosed with ASD have a higher risk for becoming overweight [163]. More recently, it has been demonstrated that APP mediates diet-dependent weight gain, probably through enhanced TNF-α and IL-6 secretion by macrophages, and the potentiation of cholesterol uptake by colonic epithelial cells [164,165,166]. Moreover, high-fat diet-induced APP production in white adipose tissue leads to mitochondrial dysfunction [167]. However, the role of ADAM10 in cleaving APP in relation to weight gain has not been studied. Taken together, it can be hypothesized that, possibly, ADAM10-induced dysregulated APP can be involved in the development of obesity in ASD.

In conclusion, ADAM10 has an important role in maintaining intestinal homeostasis. However, there are no studies conducted about the role of enhanced ADAM10 on intestinal functioning in ASD, and this will be crucial to deepening our understanding of this involvement.

### 5.3. ADAM17 in the Intestinal Tract

ADAM17 is ubiquitously expressed in all intestinal epithelial cells. In a murine model of decreased ADAM17 expression, where its activity is significantly reduced, normal intestinal epithelial cell proliferation is not compromised. However, there is a less effective response of the intestines to inflammation [168]. ADAM17 seems to be an essential component in regulating intestinal inflammation. The proinflammatory cytokines TNF-α and IL-6 [8] can disrupt the tight junction structure in the intestine and contribute to inflammation [169,170]. As ADAM17 cleaves TNF-α [171] and IL-6R [139], it seems that ADAM17 activity is tightly connected to the intestinal barrier integrity via a proinflammatory route. Recently, research pointed out that the polyphenol, resveratrol, is able to ameliorate social deficits in the VPA mouse model of ASD, probably attributing to its anti-inflammatory properties [172]. Resveratrol also reduced the proinflammatory cytokine levels, such as IL-6 and TNF-α in the BTBR T+tf/J mouse model of ASD, which indicates that the inhibition of inflammation may be promising to the treatment of ASD [173]. Additionally, with the observed increased levels of TNF-α, sIL6R/IL6 complexes [134,174] and ADAM17 [130] in ASD patients, it is tempting to assume that this metalloprotease could be crucial in the development of ASD by alteration of the gut–brain axis.

### 5.4. ADAMs and Intestinal Microbiota

Another important aspect of the gut–brain axis is the intestinal microbiome. Intestinal microbiotas consist of cohabitating microorganisms involved in regulating the host immunity and inflammation [175,176,177]. There is bidirectional communication between the intestinal microbiota and the brain [178,179,180]. Although there are limited reports on the contribution of ADAM10 or ADAM17 to intestinal microbiota dysbiosis, it is possible to connect several studies and hypothesize.

The composition of the intestinal microbiome is altered in ASD children compared to normal healthy individuals [8,9,84,151]. Their microbial composition contains a higher proportion of Gram-negative bacteria [181], which increases the expression of lipopolysaccharides (LPS) [182]. LPS can activate the innate immune system through the activation of Toll-Like Receptor 4 (TLR4) [183,184]. The activation of TLR4 stimulates the production of proinflammatory cytokines in the intestines and causes neuroinflammation in the brain by activation of the microglia [185,186]. Strikingly, the activation of TLR4 stimulates the ADAM17-dependent shedding of TNF-α [187]. Furthermore, proinflammatory cytokines and LPS treatments are both able to increase the active ADAM10 levels in vitro [188]. It can be suggested that the altered microbiome in people diagnosed with ASD might be able to activate ADAM17 and ADAM10 through increased LPS and proinflammatory cytokines production.

A higher incidence of *Clostridium perfringens* in fecal samples of ASD children has been described compared to healthy children [189,190]. In the intestines, several species of *Clostridium perfringens* generate potent toxins that are the causatives of fatal intestinal and CNS diseases in animals [191]. Delta-toxin is one of these that perturbs the intestinal epithelial barrier function in human intestinal epithelial Caco-2 cells through enhancing the ADAM10 activity in a dose- and time-dependent way, which is blocked in the presence of the ADAM10 inhibitor [192,193]. Therefore, the altered microbiome and related metabolites in ASD patients seem to be able to activate ADAM17 and ADAM10 by increasing the production of LPS and Delta-toxin. However, there is little known about the effects of other bacterial-generated metabolites on ADAM10 and ADAM17, such as p-cresol, its derivative p-cresyl sulfate (pCS) and 4-ethylphenylsulfate (4EPS). The levels of p-cresol and its conjugated derivative pCS are increased in urine and fecal samples in ASD children [97,194,195]. Urinary p-cresol has been suggested as a biomarker for ASD in small children because of its significant elevation [195]. A 4EPS treatment induced ASD-like behavior in mice [8]. Mishra et al. showed that there was no difference in the cecal bacterial microbiota composition and load between ADAM17 conditional KO mice and control mice, but the conditional knockout of ADAM17 decreased the peritoneal spread of bacteria following sepsis induction compared to control mice [196], which might be involved with reduced cleavages of TNF-α shedding and other proinflammatory cytokines by ADAM17.

Additionally, some substrates of ADAM17 or ADAM10 can regulate microbiota composition. Angiotensin-converting enzyme 2 (ACE2), a substrate of ADAM17 [197,198], plays an emerging role in the pathogenesis of cardiovascular and lung diseases through changing the composition of the intestinal microbiota, such as increasing the ratio of the Firmicutes to Bacteroidetes and decreasing the *Bifidobacterium* genus, which raises the potential relation of ADAM17-mediated ACE2 shedding and intestinal dysbiosis [199,200,201]. Meantime, it is demonstrated that an enteric infection coupled with chronic Notch receptor pathway inhibition in mice is associated with bacterial dysbiosis compared to control mice, indicated as a significant decrease in the Bacteroidetes phyla, with concomitant increases in the Firmicutes, Proteobacteria and Verrucomicrobia phyla [46]. It seems that the Notch receptor inhibition changed the microbiota composition in enteric-infected mice, which implies that ADAM10 overexpression may have a similar effect on microbiota composition through cleavage of the Notch receptor. Taken together, intestinal microbes can regulate ADAM10 or ADAM17 activity by producing bacterial metabolites, and, in turn, ADAM10 or ADAM17 can also change the intestinal microbiome composition. However, these connections remain unclear, and more research is needed.

### 5.5. ADAMs and the Immune System

The development and function of the immune system is highly dependent on the intestinal microbiota, as demonstrated by the limited immune activity in germ-free mice [202]. Intestinal bacterial fermentation produces a wide range of metabolites on the basis of tryptophan, tyrosine and phenylalanine from our daily diet, such as serotonin, short-chain fatty acids (SCFAs), indole-containing metabolites and p-cresol [203,204,205]. The role of SCFAs and other not-mentioned bacterial metabolites in ASD have been extensively reviewed [177]. These metabolites can regulate immune responses and inflammatory responses by recognizing their receptor on epithelial cells or entering into the systemic circulation or the brain. SCFAs promote the number, function and differentiation of colonic T-reg cells in mice [203,206]. In addition, SCFA—specifically, butyrate—being fuel for epithelial cells, promote intestinal barrier integrity [207,208]. In the brain, SCFAs also increase the microglia maturation and functions in mice [204]. However, the exact mechanisms how these metabolites affect host immune and brain functions remains to be investigated. In recent years, alterations in the gut–brain axis have been presented as possible pathological causes of ASD, and targeting the intestinal microbes has been recognized as a promising treatment for ASD [8,16,178,209]. For example, maternal immune activation (MIA) induced by polyinosinic:polycytidylic acid (polyI:C) injection led to intestinal dysbiosis in the male offspring associated with defects in communicative, stereotypic, anxiety-like and sensorimotor behaviors. The oral administration of *Bacteroides fragilis* restored these ASD-like symptoms [8]. The roles of ADAM10 and ADAM17 in the effects of the bacterial metabolites in the immune system in ASD has not been researched well; however, it is possible to speculate about its potential involvement.

Transforming growth factor β1 (TGF-β1) is one member of the TGFβ family and, generally, regulates T lymphocytes and antigen-presenting cells as an immunosuppressor [210]. Some studies showed that there is a significant decrease of the TGF-β1 level in the plasma or serum of ASD children [211,212]. Moreover, one study showed that TGF-β1 might be considered as a biomarker of ASD severity. Increasing TGF-β1 levels in the plasma of ASD children consequently improved the behavioral rating score [213]. TGF-β1 also is essential to the microglial development, phenotypes and functions in vitro and in vivo, which is connected with ASD pathogenesis [214,215,216]. Besides, TGF-β1 plays a vital role in modulating social interactions and repetitive behaviors in mice hippocampus. It was demonstrated that adult hippocampal TGF-β1 overexpression increases social interactions and decreases self-grooming and depression-related behaviors, and early hippocampal TGF-β1 overexpression reversely decreases those behaviors [217]. Kawasaki et al. illustrated that TGF-β1 signaling is dependent on ADAM 17 activity, and thus, modifying ADAM17 genetic variants enhances TGF-β1 signaling activity through cleaving less type 1 TGF-beta receptor (TGF-βR1) in mice and humans [218]. It is demonstrated that TGF-βR1 is a substrate of ADAM17, and its cleavage by ADAM17 downregulates TGF-β1 signaling through the decreasing cell surface TGF-βR1 [218,219]. In addition, Vasorin is a type 1 transmembrane protein, and it is cleaved by ADAM17 to generate soluble Vasorin that binds to TGF-β1 as a suppressor [220,221]. To our knowledge, there is little known about the ADAM17-mediated TGF-β1 signaling in ASD pathogenesis.

T-helper 17 lymphocytes (TH17) cells and their effector cytokines interleukin 17 (IL-17) are necessary for immune responses against extracellular bacteria and fungi, and their dysregulation is thought to underlie numerous inflammatory and autoimmune diseases, such as inflammatory bowel disease and multiple sclerosis [222]. Choi et al. demonstrated that maternal immune activation induced ASD-like behavioral phenotypes in the offspring and compromised their cortical brain development in an IL-17a-dependent manner. IL17a KO or IL-17a blockage with an antibody rescued the ASD-like phenotypes [223]. The proinflammatory cytokine IL-6 inhibits TH17 cell differentiation from the CD4+ T-cell subset as an upstream regulator of IL-17a [224,225]. Therefore, IL-6 is also necessary for maternal immune activation-induced ASD-like phenotypes in the offspring [226]. It is demonstrated that ADAM17 regulates IL-6 signaling through controlling the IL-6 receptor (IL-6R) in vitro and in vivo [227]. Horiuchi et al. reported that conditional ADAM17 KO mice have increased serum levels of IL-17 compared with control littermates, indicating that decreased ADAM17 activity is associated with a downregulation of IL-17 secretion in vivo [187], but the role of the IL-17 receptor in this process was not investigated, and the definite mechanism remains to be elucidated. It is possible that the increased IL-17 level in ADAM17 KO mice is a result from the increased cleavage of the IL-17 receptor by ADAM17 [228], but it is barely studied. These findings make it promising to investigate the potential roles of ADAM10 and ADAM17 in the pathogenesis of ASD from an immunological perspective.

## 6. Metalloproteases ADAM10 and ADAM17 as Therapeutic Targets for Autism Spectrum Disorders

The metalloproteinases ADAM10 and ADAM17 are or might be involved in different aspects of ASD pathogenesis. Strikingly, as can be concluded from Table 2, it seems that the enhanced expression and activity of ADAM10 or ADAM17 might contribute to several aspects of ASD. Therefore, the reduction or inhibition of these targets could be interesting as therapeutic strategies for ASD. Although not much attention has been given in research to targeting metalloproteases, some possible therapeutic options will be discussed below.

### 6.1. TIMPs

The natural inhibitors of the metalloproteases are the Tissue Inhibitors of Metalloproteases (TIMPs). There are four members of the TIMP family [229,230]. In general, the TIMPs can inhibit all MMPs, but the strength of MMP inhibition differs between the TIMPs. Interestingly, TIMPs inhibit ADAMs with higher specificity. For example, ADAM10 is specifically inhibited by TIMP-1 and TIMP-3 [231]. Moreover, TIMP-3 also inhibits ADAM17 [232]. The main limitation of metalloprotease inhibitors is their lack of selectivity. Therefore, an inhibitor can affect other enzymes as well, which could lead to undesirable side effects. For instance, increasing the TIMP-3 levels can be an interesting therapeutic target, as this can possibly decrease both the ADAM10 and ADAM17 activity in ASD patients; however, TIMP-3 also has an inhibitory effect on most MMPs [233,234]. Thus, the search for selective inhibitors is of critical importance in order to be used as a therapeutic drug.

### 6.2. ADAM Inhibitors

In search for molecules with a great selectivity for ADAM10 and ADAM17, the GI254023X compound has been identified as a potent and selective inhibitor of ADAM10, with 100-fold higher selectivity than ADAM17 [235,236]. In addition, Mahasenan et al. recently synthesized and tested the compound (1*R*,3*S*,4*S*)-3-(hydroxycarbamoyl)-4-(4-phenylpiperidine-1-carbonyl) cyclohexyl pyrrolidine-1-carboxylate, which showed a high potency of inhibiting ADAM10 [237]. This compound can also cross the BBB [237]. Furthermore, Hirata et al. showed that the inhibitor KP-457 has over 50 times higher selectivity for ADAM17 than ADAM10 or any other MMPs [238].

TspanC8 members regulate ADAM10 maturation and substrate selectivity. Six TspanC8 members can form six different Tspan–ADAM10 complexes, which preferentially cleave different substrates as six scissors [36,62,65,125]. Therefore, the development of inhibitors targeting these complexes is beneficial compared to the side effects of broad ADAM inhibitors. Taken together, these studies showed that effective inhibitors are available and might be beneficial for ASD treatment.

### 6.3. Probiotics, Bacterial Metabolites and Prebiotics?

As discussed above, intestinal dysbiosis has been frequently described in children suffering from ASD. Pro-, prebiotic and even microbiota transfer therapy (MTT) interventions have been proposed as promising treatments for ASD children [16,239,240,241,242]. Of interest are the bacterial metabolites 4EPS, as well bacterial toxin LPS, that induce ASD-like behavior in mice by unknown mechanisms [8,243]. Furthermore, a bacterial LPS-induced increase of ADAM10 expression is important for proinflammatory immune cell responses [244]. Other important bacterial metabolites are SCFAs. The increase of enteric SCFAs levels in ASD mice [245], as well as in ASD children, are demonstrated [246,247]. The precise mechanism of action of SCFAs in relation to ASD-like behaviors is not known, but the effects on the mitochondrial function or epigenetic alterations in the brain may be involved [248]. Given the important roles of ADAM10 and ADAM17 in the gut functions, immunity and brain, it will be interesting to study whether and how intestinal microbiota-derived metabolites, such as 4EPS, LPS, p-cresol and SCFAs, affect ADAM10/17 activity in the intestinal tract and in the brain related to ASD. When, indeed, an important role of these bacterial metabolites on the ADAM10/17 activity is established, then targeting the intestinal microbiota with pre- and probiotics, as well as MTT, may be useful in ASD.

### 6.4. Targeting ADAM10 and ADAM17 in ASD: Some Considerations

The strategy of targeting ADAM 10 and/or ADAM17 as future treatments in ASD raises several issues. First, the processes in which ADAMs are involved are critical for cell, tissue and organ functioning; therefore, ADAM10 and ADAM17 inhibitors might have serious side effects. Both proteases contribute to developmental and regenerative processes; for example, the disruption of ADAM17 in mice leads to death, and studies in KO mice show that ADAM10 is vital for early development [125]. The specific targeting of ADAM10 or ADAM17 at the right time and in the right location might be the way to go. Secondly, it should be investigated at which location ADAM10 and ADAM17 should be targeted: in the intestines or the brain. For the latter, compounds that are able to pass the BBB are essential. Indirect targeting through manipulation of the intestinal microbiota with pre-, pro or postbiotics might be a safer way to inhibit the enhanced ADAM10 and ADAM 17 activity in ASD.

## 7. Outlook and Conclusions

ASD is a highly heterogeneous disorder that includes multiple affected genes, altered synaptic density, neuroinflammation, low-grade systemic immune activation and an intestinal phenotype, including a “leaky gut”. Therefore, it is challenging to pinpoint what the exact underlying cause of this neurodevelopmental disease is. The ASD-associated enhanced expression and/or activity of the metalloproteases ADAM10 and ADAM17 provide an overarching hypothesis that affects many different aspects that seem to be involved, at least in part, in ASD pathology. ADAM10 is responsible for the proteolytic cleavage of several key proteins involved in synapse formation, axon signaling and cell adhesion and for regulating the intestinal permeability. Furthermore, ADAM17 has a pivotal role in the shedding of proteins that regulate the onset of (neuro)inflammation and immune responses. Additionally, the effects of ADAM10 and ADAM17 on the intestinal microbiota composition and the effects of bacterial metabolites on ADAM10 and ADAM17 expression and activity remain to be investigated. Taken together, these two metalloproteases seem responsible for activating key pathways that seem to be altered in ASD pathogenesis. Figure 3 provides an overview of the pathways where ADAM10 and ADAM17 are possibly involved in the pathogenesis of ASD. In conclusion, the enhanced expression and/or activity of ADAM10 or ADAM17 could possibly be involved in the induction and maintenance of ASD-like phenotypes in the brain, as well as systemically and in the intestinal tract. Consequently, this hypothesis suggests that reducing the levels or activity of ADAM10 or ADAM17 could be a potential therapeutic target in ASD patients.

In order to provide more evidence to support this hypothesis, it is necessary to further determine if there is any altered ADAM10 and ADAM17 expression and/or activity in ASD-associated mouse models or in ASD patients. Moreover, more studies need to be conducted to investigate the role and molecular mechanisms of ADAM10 and ADAM17, which will shed light on the molecular pathogenesis and possible targets for the treatment of ASD. Additionally, it will be interesting to screen metalloprotease-specific inhibitors and then test the specific inhibitors of ADAM10 and ADAM17 in ASD animal models.

## Figures and Tables

**Figure 1 ijms-22-00118-f001:**
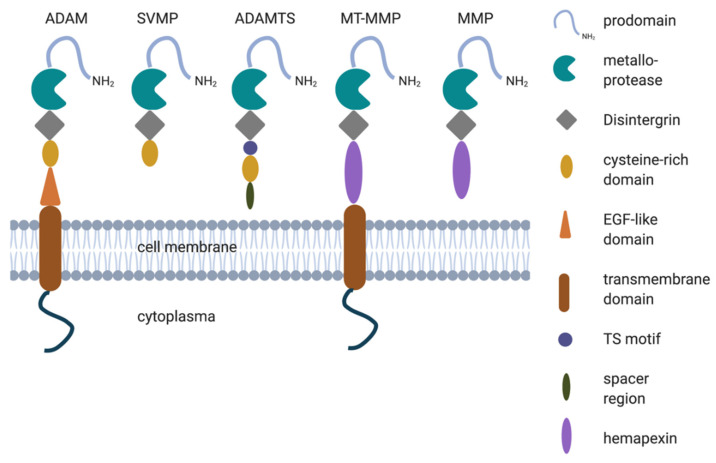
Protein structure of the members of the metzincin family of metalloproteases: A Disintegrin And Metalloprotease (ADAM), Snake Venom Metalloprotease (SVMP), A Disintegrin And Metalloprotease Thrombospondin motif (ADAMTS), membrane-type matrix metalloproteinases (MT-MMP) and MMP.

**Figure 2 ijms-22-00118-f002:**
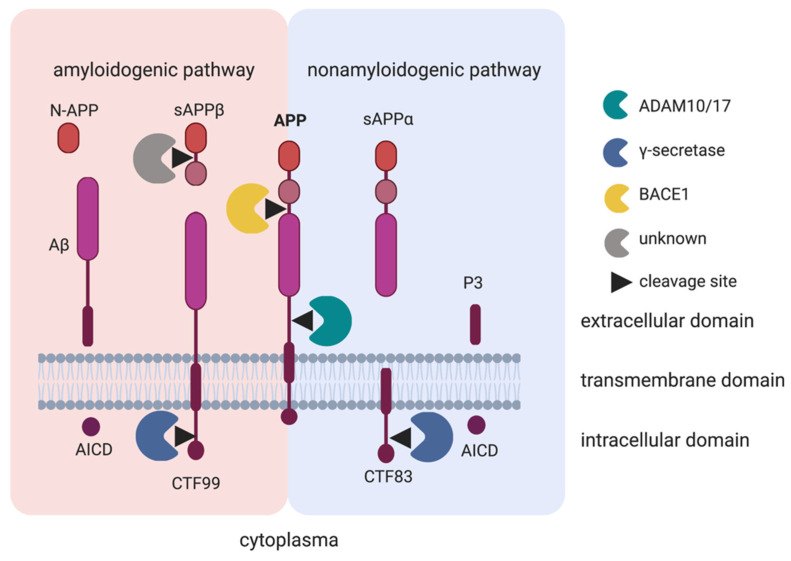
Overview of cleavage products of the Amyloid Precursor Protein (APP) by different secretases. The cleavage of APP by the β-site APP cleaving enzyme 1 (BACE1) initiates the amyloidogenic pathway, where the soluble fraction of APP (sAPP)β and C-Terminal Fragment 99 (CTF99) are created. γ-secretases further process the CTF99 to create the neurotoxic Aβ protein and the Amyloid Precursor Intracellular Domain (AICD). Meantime, sAPPβ is also cleaved by additional secretases at an unknown site to generate an N-terminal of APP (N-APP). The cleavage of APP by either ADAM10 or ADAM17 initiates the nonamyloidogenic pathway, which creates sAPPα and C-Terminal Fragment 83 (CTF83). γ-secretases then cleave CTF83 to create P3 and AICD.

**Figure 3 ijms-22-00118-f003:**
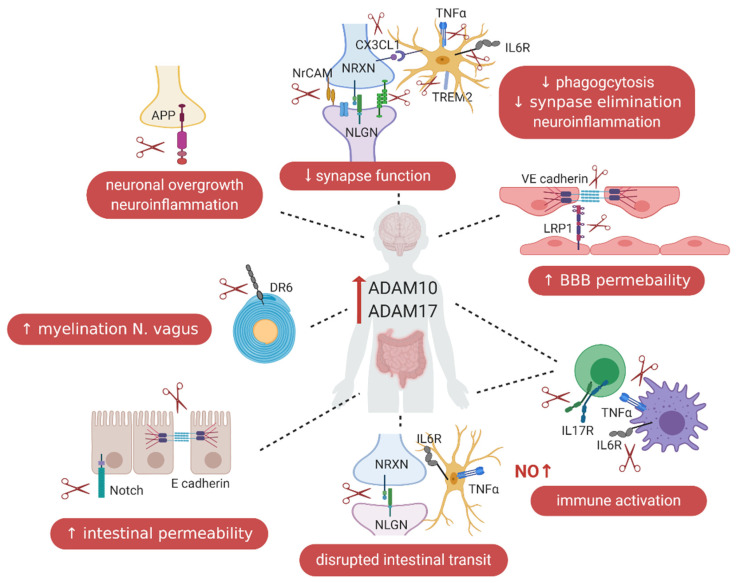
Overview of the proposed role of ADAM10 and ADAM17 in ASD pathology, with a focus on the gut–immune–brain axis. In the intestines, the increased activity or levels of ADAM10 and ADAM17 will lead to an increased intestinal permeability by cleaving more E-cadherin and Notch, increased intestinal inflammation by cleaving more IL-6R and TNF-α, disrupted intestinal transit by cleaving more NLGN3 and increased myelination of nervus vagus by cleaving more DR6 in the Enteric Nervous System; these contribute to the intestinal dysfunctions. In the brain, the elevated activity of ADAM10 and ADAM17 will result in increased neuronal growth, synaptic density, dendritic spines densities and larger brain volume by cleaving more synaptic molecules, such as NLGN, NRXN, NrCAM and APP. Furthermore, it will result in decreased synaptic elimination and microglial phagocytosis by cleaving more CX3CL1 and TREM2; increased neuroinflammation by cleaving more IL-6R, TNF-α and TREM2 and increased BBB permeability by cleaving more VE-cadherin and LRP1. These processes will lead to altered brain development and functions. Additionally, increased ADAM10 and ADAM17 activity will lead to immune activation by cleaving more IL-17R, IL-6R and TNF-α. All of these can participate ASD development and be involved in ASD pathogenesis.

**Table 1 ijms-22-00118-t001:** The reduction in shedding of the A Disintegrin And Metalloprotease (ADAM)10 substrates after the conditional deletion of ADAM10 in embryonic primary cortical neuron cultures [35]. Abbreviations: Fractalkine (Cx3cl1), Neuroligin-1 (NLGN-1), Protocadherin-9 (PCDH9), Neural glial-related Cell Adhesion Molecule (NrCAM), Neuroligin-3 (NLGN-3) and Amyloid-β Precursor Protein (APP).

ADAM10 Substrates	Shedding Reduction in ADAM10^–/–^ Neurons
Cx3cl1	91%
NLGN-1	83%
PCDH9	71%
NrCAM	66%
NLGN-3	62%
APP	20%

**Table 2 ijms-22-00118-t002:** The effect of ADAM10 and ADAM17 on different substrates and their involvement in Autism Spectrum Disorder (ASD) patients. ↓: downregulated; ↑: upregulated

Protein Name	Gene Symbol	ADAM10 Shedding	ADAM17 Shedding	ASD Patients
Amyloid Precursor Protein	*APP*	↑sAPPα [32]	↑sAPPα [32]	↑ sAPPα [28]
Neuroligin-1	*NLGN-1*	↓ Synaptogenic activity [92]		Common variants [249,250]
Neuroligin-3	*NLGN-3*	↓ Synaptogenic activity [92]		R415C transition [251,252] Common variants [250]
Neurexin-1	*NRXN-1*	↓ Synaptogenic activity [93]		Loss-of-function variants [24,26,27]
Neural glial-related Cell Adhesion Molecule	*NrCAM*	↑ Axon targeting activity [35]		SNPs & Common variants [25]
Protocadherin9	*PCDH9*	No data available		Copy Number Variants [24]
Fractalkine	*CX3CL1*	↑ Synaptic pruning [35,121,122]		No data available
Tumor Necrosis Factor-α	*TNF-α*		↑ pro-inflammatory activity [131]	↑ in blood and brain [134]
Interleukin-6 Receptor	*IL-6R*	↑ pro-inflammatory pathways [139]	↑ pro-inflammatory pathways [139]	↑ IL-6 in blood and brain [134]
Triggering Receptor Expressed in Myeloid cells 2	*TREM2*		↓ TREM2 membrane receptor levels [143]	↓ in post-mortem brain tissue age 5–23 [145]

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
