# Peer review of "The Gut-Brain Axis in Autism Spectrum Disorder: A Focus on the Metalloproteases ADAM10 and ADAM17"

_ijms, 2020, doi:10.3390/ijms22010118_

Round 1

Reviewer 1 Report

Dear Authors,

I have read your review manuscript "The gut-brain axis in Autism Spectrum Disorder: A focus on the metalloproteases ADAM10 and ADAM17" which addresses a potential connection between the metalloproteases ADAM10 and ADAM17, and development of autism spectrum disorder (ASD). The manuscript is well written, interesting, and provides broad insight into the potential role of these enzymes in ASD. Also, the graphic material is well done.

I have few comments which would, I my opinion, improve the manuscript:

  1. I somehow miss a short description of the other MMPs (MT-MMPs, ADAMTs and SVMPs) – why or why not are they connected to ASD? A short comment on this, at lines 106-107 would be valuable for the reader.
  2. The process of sequential cleavages depicted in figure 2 would be clearer if the arrows were put lower and would show where enzymes cleave, while for the distinction between the two pathways a simple vertical line would suffice. This is merely a design issue.
  3. There is something wrong with the text at the end of abstract (line 27)...

A more substantial concern is connected to targeting ADAMs using inhibitors seems, as mentioned in section 6. Such strategy raises several alerts, also with regard to inhibitor specificity, localized targeting etc. Also, the processes in which ADAMs are involved are critical for normal cell/tissue/organ functioning. I believe a more detailed addressing of these issues, also describing possible side effects, should be presented.

To conclude, the authors did a good job at presenting the current knowledge on the rather novel connection between ASD and ADAMs. Therefore, I would recommend aaccepting the manuscript for publication providing the authors address the abovementioned issues.

Author Response

We thank this reviewer for the appreciation of our manuscript and for the fruitful comments. Please, see attachment with a point-by-point reply. In addition, we have changed the manuscript accordingly.

Reviewer 2 Report

Very interesting review on the possible importance of ADAMs in the pathogenesis of ASD. The authors thoroughly described the role of ADAM10 and ADAM 17 in the functioning of the nervous and immune systems and the gut-brain axis.

However, I have some minor comments and suggestions:

  1. in the abstract - line 20-21 - incomprehensible meaning of the sentence, especially "all mechanisms that are involved in ASD pathology as well", this should be reworded
  2. Authors should standardize the description of the ADAM abbreviation - once it is A Disintegrin And Metalloproteins, once it is "and" in lowercase

3.The abbreviations entered are not used in the text (e.g. NLGNs and NRXNs, line 54, no abbreviation line 69, KO line 266, 268) or vice versa are given abbreviations which are then not used (VPA, line 344; ADI-r , line 389)

  1. Table 1 - Authors should explain the origin of ADAM10 - / - neurons
  2. Line 191 –Authors wrote " several groups of ASD-affected individuals " which groups? age? severity?
  3. There are abbreviations for bacteria that were not explained before, e.g. L. reuteri - line 181, B. fragilis - line 539
  4. Line 181 - Too large a mental abbreviation that should be explained - is it about supplementing mice with the L. reuteri strain? (what strain?)
  5. Line 299 - Authors wrote that "Insufficient synaptic elimination ....... is related to the macrocephaly ......" Is it a confirmed relationship?, or it can it be related

9.Line 339 – There is the sentence: "There are other proteases that are able of ....... cleaving TNF-alpha" Could the Authors specify any other proteases?

  1. Line 486 – “TLR4 stimulates…….” rather activation of TLR-4 by different bacterial metabolites ot components stimulates pathways………
  2. Line 531 – SCFAs promote the size…….. Size or rather numbers of Treg?
  3. I missed the information in the manuscript that SCFAs are also of great importance in the proliferation of epithelial cells and the formation of the epithelial barrier, which plays a role in the functioning of the intestinal barrier and the gut-brain axis. Authors should supplement the review with this information. Line 618 - The authors write about the mechanism of action of SCFA, but also do not mention the possible effect on the epithelial barrier.

Author Response

We thank this reviewer for the appreciation of our manuscript and for the fruitful comments. Please, see attachment with a point-by-point reply. In addition, we have changed the manuscript accordingly

Reviewer 3 Report

Accepted

Author Response

We thank the reviewer for the advice to accept our manuscript for publication